# Conformational Disorder Analysis of the Conditionally Disordered Protein CP12 from *Arabidopsis thaliana* in Its Different Redox States

**DOI:** 10.3390/ijms24119308

**Published:** 2023-05-26

**Authors:** Alessandra Del Giudice, Libero Gurrieri, Luciano Galantini, Silvia Fanti, Paolo Trost, Francesca Sparla, Simona Fermani

**Affiliations:** 1Department of Chemistry, Sapienza University of Rome, 00185 Rome, Italy; alessandra.delgiudice@uniroma1.it (A.D.G.); luciano.galantini@uniroma1.it (L.G.); 2Department of Pharmacy and Biotechnology, University of Bologna, 40126 Bologna, Italy; libero.gurrieri2@unibo.it (L.G.); paolo.trost@unibo.it (P.T.); 3Department of Chemistry “G. Ciamician”, University of Bologna, 40126 Bologna, Italy; silvia.fanti5@unibo.it (S.F.); simona.fermani@unibo.it (S.F.); 4Interdepartmental Centre for Industrial Research Health Sciences & Technologies, University of Bologna, 40064 Bologna, Italy

**Keywords:** small angle X-ray scattering (SAXS), intrinsically disordered protein, conditionally disordered protein, post-translational modification, redox regulation, structural characterization, ensemble optimization, unstructured biology

## Abstract

CP12 is a redox-dependent conditionally disordered protein universally distributed in oxygenic photosynthetic organisms. It is primarily known as a light-dependent redox switch regulating the reductive step of the metabolic phase of photosynthesis. In the present study, a small angle X-ray scattering (SAXS) analysis of recombinant Arabidopsis CP12 (AtCP12) in a reduced and oxidized form confirmed the highly disordered nature of this regulatory protein. However, it clearly pointed out a decrease in the average size and a lower level of conformational disorder upon oxidation. We compared the experimental data with the theoretical profiles of pools of conformers generated with different assumptions and show that the reduced form is fully disordered, whereas the oxidized form is better described by conformers comprising both the circular motif around the C-terminal disulfide bond detected in previous structural analysis and the N-terminal disulfide bond. Despite the fact that disulfide bridges are usually thought to confer rigidity to protein structures, in the oxidized AtCP12, their presence coexists with a disordered nature. Our results rule out the existence of significant amounts of structured and compact conformations of free AtCP12 in a solution, even in its oxidized form, thereby highlighting the importance of recruiting partner proteins to complete its structured final folding.

## 1. Introduction

Chloroplast protein 12 (CP12) was first discovered in land plants and is characterized for its interaction with photosynthetic glyceraldehyde-3-phosphate dehydrogenase (GAPDH; EC 1.2.1.13) [1]. Later, CP12 was reported to be a small intrinsically disordered protein (IDP), conserved among most oxygenic photosynthetic organisms from cyanobacteria to higher plants [2,3,4,5]. Nowadays, CP12 is considered a redox-dependent conditionally disordered protein (CDP) since it has been reported that the formation of two regulatory disulfide bridges induces a disorder-to-order transition [6,7,8].

Based on primary structure, eight different groups of CP12 are recognized [5]. Canonical CP12s are proteins of about 80 amino acids, containing an N-terminal Cys pair, followed by the core sequence AWD_VEEL and ending with a second Cys pair at the C-terminus of the protein. Of the other seven groups, four are variants of the canonical CP12 (e.g., one or both Cys pairs as well as the core sequence can be missing), and three are fused at the N-terminus with a cystathionine-β-synthase (CBS) domain [5]. CBS-fused CP12s are more abundant in cyanobacteria than in algae and plants [5,9].

Phylogenetic analyses of CP12s do not recognize any of the eight groups deriving from primary structural features. The phylogenetic distribution of CP12 types suggests that all photosynthetic organisms have at least one CP12, which is assumed to derive from very ancient duplication, divergence and fusion events of ancestral CP12 [3,4,5,10].

After the first evidence of the existence in spinach leaves of a CP12-dependent complex containing both GAPDH and phosphoribulokinase (PRK, EC 2.7.1.19), two energy-consuming enzymes of the Calvin–Benson cycle [11], regulatory complexes formed by GAPDH-CP12-PRK have also been found in many other land plants [12,13], algal species [14,15] and cyanobacteria [16,17,18]. Although the stoichiometry of the regulatory ternary complex has not been firmly established for all identified supramolecular complexes, it is generally accepted that four CP12s bind two GAPDHs and two PRKs (GAPDH_2_-CP12_4_-PRK_2_). In the complex, both GAPDH and PRK activities are negatively affected, in agreement with the conditions that drive the complex formation in dark-adapted chloroplast stroma. In fact, while reduced thioredoxins, NADP(H) and the GAPDH substrate 1,3-bisphosphoglycerate (BPGA) de-stabilize the GAPDH_2_-CP12_4_-PRK_2_ complex, oxidized conditions and NAD(H) do the opposite [2,19]. In addition to the well-defined role of a scaffold in assembling the ternary complex with GAPDH and PRK, various studies have assigned other functions to CP12, which has earned it the nickname “Jack of all trades” [20,21,22,23,24].

Recently, the 3D structures of the inhibited GAPDH_2_-CP12_4_-PRK_2_ complex, from both the land plant *Arabidopsis thaliana* and the thermophilic cyanobacterium *Thermosynechococcus elongatus*, have been solved, and the structure of CP12 embedded into the ternary complex is the only high-resolution structure available of the entire protein [17,25]. Locked within the ternary complex, the four CP12s show a similar overall structure. Two N-terminal parallel α-helices form the PRK binding domain, which is connected via a flexible linker to the C-terminal GAPDH binding domain. Each of the two domains contains one pair of redox-responding cysteines, each constraining the CP12 structure by forming disulfide bonds [25]. Even within the complex, the behavior of CP12 is peculiar in being able to promote different compatible interactions between GAPDH and PRK, apparently because of the ability of CP12 to explore different angles of the surrounding space via the rotation of the flexible linker [17,25]. The CP12 C-terminal GAPDH binding domain is largely devoid of secondary structures and includes a tail that can fully occupy one active site of GAPDH [17,25].

Outside the complex, different biophysical approaches have contributed to demonstrate that reduced CP12 clearly belongs to the group of IDPs [4,26,27,28], while, upon oxidation, it shows a less disordered structure, although one that is not as compact as the formation of two disulfide bonds might suggest. Recently, a synergistic combination of biophysical techniques shed light on the possible conformers assumed by *Chlamydomonas reinhardtii* CP12 (CrCP12) in its oxidized state [7]. These data indicate the coexistence of two main ensembles of conformers. The slightly more abundant conformational ensemble (≈60%) describes a protein organized as follows: (i) two long α-helices with opposite orientations (Leu8-Ala24 and Ser28-Asp50) face each other and are connected by a hairpin containing the N-terminal disulfide bond occurring between Cys23 and Cys31; (ii) the C-terminal region (Leu62-Tyr78) with the second disulfide bridge (Cys66-Cys75) and two short α-helices formed after oxidation. In the less abundant conformational ensemble (≈40%), the C-terminal region maintains the same elements of a secondary structure, while the N-terminal region shifts from an ordered to a disordered organization highlighting its high dynamism [7,8]. Based on this evidence, the CrCP12 has been defined as a conditionally disordered protein, rather than an intrinsically disordered one [6,7].

Small angle X-ray scattering (SAXS) allows the study of protein conformations in a solution. In particular, it is a suitable tool for the structural characterization of disordered proteins and/or dynamic systems. SAXS analysis helps to interpret the apparent disorder in terms of populations of different conformers [29]. In the present study, the SAXS approach was used to disclose Arabidopsis CP12 structural ensembles under different redox states. Differences found between the behavior of CP12 in higher plants (here, *Arabidopsis thaliana*) and single-cell green algae (*Chlamydomonas reinhardtii* [7]) are also discussed.

## 2. Results

### 2.1. AtCP12 in Solution Is Disordered and More Expanded in Reduced Form Compared to Its Oxidized Form

Fully oxidized AtCP12 and fully reduced AtCP12 were prepared as described in Section 4.2 according to [4,30,31]. The superimposition of the SAXS profiles of oxidized and reduced AtCP12 shows some differences between the two forms. Specifically, the oxidized form appears more compact compared to the reduced form (Figure 1a). Moreover, the comparison in the dimensionless Kratky plot (Figure 1b) between AtCP12 profiles and the profiles of a fully folded (BSA) and an unfolded protein (SASBDB entry SASDNP6 [32]) as reference, highlights that in either redox condition AtCP12 is largely disordered and displays the typical behavior of a highly flexible biomolecule.

The Kratky plot is a useful representation to monitor the folding state of proteins in a solution. Globular particles show a bell-shaped profile with a well-defined maximum, while particles with the features of a flexible polymer produce a Kratky plot showing a plateau or a linear growth when local rod-like rigidity is present [33]. A dimensionless version, obtained by normalizing the intensity by its zero-angle value and by multiplying the q range by the R_g_ of the overall particle, has been proposed as a useful tool to compare the flexibility level of different proteins and to recognize the co-presence of folded and disordered domains [34].

The R_g_ values obtained by the Guinier linear fit (2.3 nm reduced vs. 2.0 nm oxidized) and from P(r) function integration (Figure 1c,d and Appendix A) confirm that reduced AtCP12 adopts a more expanded conformation compared to the oxidized form. The P(r) function, which can be interpreted as a histogram of the internal distances within the protein particles, shows that, for oxidized AtCP12, the maximum shifted to lower distances and that the maximum width was 7 ± 2 nm narrower than that of the reduced form. In the reduced form, the P(r) shows significantly high values for distances below 5 nm, while it approaches zero at distances larger than 7 nm, suggesting that, in reduced conditions, expanded conformations are significantly more populated than in oxidized ones, affecting the average scattering profile. The expected R_g_ value for a disordered protein in an aqueous solution with the same length of AtCP12 can be estimated according to a Flory-equation parameterization in agreement with random-coil polymer behavior (*R*_g_ = *R*_0_*N^v^*, where *R*_0_ = 0.254 ± 0.001, *v* = 0.522 ± 0.01, and *N* is the number of monomers [35]) and results in 2.35 nm as a lower boundary. Comparing the experimental R_g_ values, we observe that reduced AtCP12 agrees with this prediction, whereas the R_g_ of oxidized AtCP12 lies beneath the Flory-region of IDPs, indicating that the protein is slightly more compact than a total random-coil-like peptide.

### 2.2. Estimate of the Contribution from Different Structural Ensembles in the Solution States of AtCP12

In order to find the optimal ensemble of conformers fitting the AtCP12 experimental scattering profiles in the two redox states, we built structural pools of possible AtCP12 conformers and applied the ensemble optimization method (EOM) [36,37]. Five different pools of about 10,000 conformers were generated (Figure 2a), with increasing levels of structural order, as assessed from the distributions of R_g_ and D_max_ obtained from their calculated scattering profiles (Figure 2b,c). For brevity, the five pools are identified with three-letter names (Figure 2a).

In the *dis* pool, AtCP12 was simply represented as a 78-residue chain of amino acids with dihedral angles obeying a quasi-Ramachandran database for disordered conformations (for intrinsically disordered proteins and unstructured regions) [36].

In the *bin* pool, the C-terminal region was assumed to include the structured region of oxidized AtCP12 encompassing residues from Asp58 to Asn78 observed in the crystal structure GAPDH-CP12 binary complex (PDB ID 3QV1; [38]). This structured region is characterized by a circular structural motif stabilized by the C-terminal disulfide between Cys64 and Cys73 and comprises a short helical segment (Pro59-Asp66), a loop made of two consecutive β-turns called loop-C (Asn67-Arg74) and a short extended terminal portion named C-tail (Thr75-Asn78). The remaining N-terminal portion of the sequence (from Ala1 to Ser57) was modeled as a disordered chain, as already described for the *dis* pool.

In the *bis* pool, the assumption used for the *bin* pool was integrated with a distance constraint between Cys22 and Cys31 in order for their C_α_ atoms to be close enough (distance < 0.6 nm) to form the N-terminal disulfide. For this purpose, the atomic coordinates of the involved cysteines had to be included.

As a maximum level of structural order, the *ter* pool assumed the AtCP12 coordinates as found in the crystal structure of the ternary GAPDH-CP12-PRK complex (PDB ID 6KEZ; [25]). This highly structured conformation of oxidized AtCP12, in addition to the C-terminal circular motif that is very similar to the one observed in the binary complex, entails a full structuring of the N-terminal domain upon PRK binding. It involves a first helix from Gly6 to Thr21, a hairpin comprising the disulfide bridge between Cys22 and Cys31 and a very long helix from Val32 to Ser57. This assumption leaves, as possibly disordered residues, only the first four N-terminal residues and is clearly not in agreement with the preliminary evaluation of the flexibility of free AtCP12 in both reduced and oxidized states (Figure 1) but has been treated in the fitting attempt as a pool of 10,000 identical structures (see black lines in Figure 2b,c) for consistency and comparison with the other structural assumptions.

Moreover, the *tef* pool was built to consider a structuring level of AtCP12 intermediate between those observed in the binary complex [38] and in the ternary complex [25] crystal structures and to evaluate the consistency with structural findings reported for CrCP12 [7,39]. This pool comprises a structured N-terminal region according to the high-resolution coordinates of CP12 within the ternary complex (PDB ID 6KEZ, [25]) encompassing residues Gly6-His47, which are connected to the structured C-terminal starting from Asp57 via a 10-residue flexible linker modeled as disordered chain.

In agreement with the interpretation of the conformational behavior of CrCP12, in which the N-terminal region can fluctuate between disordered conformations and a folded helical structure [7], we considered a joint *bis–tef* pool of 20,000 conformers including both the more disordered conformers of the *bis* pool and the structures generated by the *tef* assumption, i.e., a folded helical N-terminal.

The optimization algorithm was run on all five pools separately (Figure 2) and on the additional joint *bis–tef* pool and the results were compared to find the ensemble of conformers that better describes the SAXS scattering profiles of reduced and oxidized AtCP12 in a solution. This can be assessed by evaluating the quality of the fit (from the relative values of χ^2^ and the randomness of the normalized residuals) and by comparing the R_g_ and D_max_ distributions of the starting random pool and of the selected conformers, both visually and from average values and flexibility metrics.

We observed that each of the three starting pools with a disordered N-terminal region (*dis*, *bin* and *bis*) reproduce reasonably well (χ^2^ = 0.66) the SAXS profile of reduced AtCP12. However, conformers with an R_g_ larger than 3 nm and a D_max_ larger than 9 nm were slightly oversampled compared to the random pools to fit the data, suggesting a further expanded state compared to totally random conformation (Appendix A). On the contrary, the fully disordered pool (*dis*) does not provide a comparably good fit for the SAXS data of oxidized AtCP12 (χ^2^ = 0.85). This further demonstrates that a totally random-like chain is not representative of oxidized AtCP12 in a solution, whereas the introduction of the small structural motif at the C-terminus (*bin* pool) and of the disulfide at the N-terminus (*bis* pool) improves the agreement between the theoretical scattering curve and the experimental one (χ^2^ decreases from 0.85 to 0.66; Appendix A). A further slight improvement of the fitting between the experimental and calculated profiles was gained considering the structuring in the N-terminus (pools *tef* and *bis–tef*; χ^2^ = 0.63; Appendix A).

When considering as starting possible conformers the combination of the *bis* and *tef* pools (*bis–tef* pool; Figure 3a), a quite similar and good fit to the SAXS data was possible for AtCP12 in both reduced and oxidized forms (χ^2^ = 0.68 and 0.63, respectively; Appendix A). The results of these optimizations can represent a clear way to compare the two redox states and investigate the existence of more compact conformations in the oxidized state due to a higher degree of structuring at the N-terminus.

Even if the overall R_g_ distribution obtained for the combined pool *bis–tef* shows a single maximum and a tail (Figure 3b), rather than being bimodal, it is possible to extract the weight factors to apply to the two known individual distributions of the *bis* and *tef* pools, represented as Log-normal functions, to obtain the overall distribution. The distribution of the starting *bis–tef* pool is obtained considering weight factors of 50% for both the *bis* and the *tef* pool (Figure 3b). By deconvolving in the same way the R_g_ distributions selected after fitting the SAXS data of oxidized AtCP12 (Figure 3c), we can estimate that the relative weight of the pool with a structured N-terminal (*tef* pool) would only be 25%, while 75% would be the weight of the pool with a disordered N-terminal (*bis* pool). In contrast, in the case of reduced AtCP12, the best fitting is obtained when the relative weight of the *tef* pool is negligible (Figure 3d).

## 3. Discussion

The structural characterization of naturally disordered proteins or protein regions is a challenging task because of the flexibility and dynamism of the system itself. Combining different approaches, the CP12 of the green unicellular algae *Chlamydomonas reinhardtii* has been investigated revealing that CrCP12 in the oxidized form oscillates between helical and unstructured conformers [7,8]. AtCP12 and CrCP12 share about 50% of sequence identity (Appendix A).

In this study, a SAXS approach has been selected to investigate the structural behavior of reduced and oxidized AtCP12 in a solution and compare the disorder and the dynamism of this protein with CrCP12.

The Guinier linear fit and the P(r) function integration describe an oxidized AtCP12 as more compact compared to the reduced form (Figure 1, Appendix A). However, although reduced AtCP12 confirms its behavior as a completely structureless protein, the comparison of the experimental R_g_ values obtained for oxidized AtCP12 (Appendix A) with that calculated with a Flory-equation parameterization considering a random-coil polymer shows that oxidized AtCP12 cannot be correctly described as a totally random-coil-like peptide. These results agree with the previous findings from circular dichroism (CD) and NMR spectroscopy, showing that AtCP12 in reducing conditions behaves as a typical intrinsically disordered protein, while oxidation seems to induce a slight increase in the α-helix content [4,30]. NMR analysis indicated that free oxidized AtCP12 contains a brief α-helical segment encompassing residues Pro59 to Asp66, a loop (Asn67–Cys73) showing a helicoidal structure formed by two consecutive β-turns and a C-terminal tail composed of five highly dynamic residues [38].

With the aim of identifying the optimal ensemble of conformers fitting the AtCP12 experimental scattering profiles in the two redox states and thus describing the structural behavior of this protein in a solution, we generated six structural pools of possible AtCP12 conformers (Figure 2) that were analyzed by applying the EOM procedure. The results of such analyses show that reduced AtCP12 can be well described by an ensemble composed of 100% *bis* pool, although, in general, the scattering data of reduced AtCP12 fit well with all the pools presenting a disordered N-terminal region (*dis*, *bin* and *bis* pools; Appendix A). On the contrary, oxidized AtCP12 is better described by an ensemble of conformers composed of 25% *tef* and 75% *bis* pools, indicating that, despite the presence of two disulfide bridges, AtCP12 remains largely unstructured.

The comparison of the findings obtained in the present study with those resulting from a similar approach applied to CrCP12 [7] shows that, in the oxidized state, the relative weight of conformers with a folded helical N-terminal segment (like the *tef* pool in the present work) is clearly different (60% vs. 25% in Chlamydomonas and Arabidopsis, respectively). This is not surprising considering the limited homology between the two proteins in the N-terminal region, such that CrCP12 was predicted in silico to contain largely structured portions, whereas AtCP12 was predicted as largely disordered (29% vs. 88% disordered fraction, respectively) [4].

An increase in α-helix content in oxidized AtCP12 was observed by CD in the presence of 2,2,2-trifluoroethanol [4], highlighting a certain propensity to adopt helical conformation [40]. Accordingly, the crystal structure of oxidized AtCP12 embedded in the ternary complex (GAPDH_2_-CP12_4_-PRK_2_) shows that the N-terminal domain folds in a hairpin structure formed by an α-helix (N-helix; residues Gly6 to Thr21) and a long central α-helix (residues Glu30 to Gly56) stabilized by a disulfide bond between Cys22 and Cys31 [25].

The C-terminal region of free oxidized CrCP12 shows a stable conformation resembling that assumed by AtCP12 in the GAPDH-CP12_2_ binary complex crystal structure [7,38]. Conversely, the oxidized AtCP12 C-terminal portion is characterized by a structural ensemble of 20 NMR models differing mainly for the reciprocal orientation of the helix and the following loop [38]. In the formation of the binary complex (GAPDH-CP12_2_), GAPDH exerts a selection over the conformations of the C-terminal portion of the oxidized AtCP12, inducing the synergic folding upon the binding of the C-tail in the enzyme active site [38]. The remaining N-terminal portion of AtCP12 that was not embedded in the binary complex investigated by SAXS seems to prefer compact conformations while still being highly flexible [41].

In conclusion, this study fills the information gap related to the structural properties of the N-terminal domain of free oxidized AtCP12. While in the reduced state both AtCP12 and CrCP12 are fully disordered, in oxidized conditions, despite the two disulfides formation, AtCP12 is mainly represented by unstructured conformers, unlike CrCP12, which is relatively more ordered.

## 4. Materials and Methods

### 4.1. Protein Expression and Purification

The cDNA coding for the mature form of the isoform 2 of *Arabidopsis thaliana* CP12 (gene ID At3g62410) was cloned into the pET28 expression vector (Novagen, Madison, WI, USA) in frame with an N-terminal 6XHis tag, followed by a thrombin cleavage site. Heterologous expression was performed in *E. coli* cells, strain BL21(DE3), harboring the pET28-AtCP12 plasmid. Cell growth, induction of recombinant protein and purification were carried out as in [19].

Sample purity was controlled by 12.5% SDS-PAGE, followed by Coomassie staining. Protein concentration was calculated by absorbance at 280 nm using a NanoDrop Spectrophotometer (Thermo Fisher Scientific, Waltham, MA, USA) with an average extinction coefficient of 8605 M^−1^ cm^−1^ (for both recombinant CP12s with and without the 6xHis-Tag) and a molecular mass of 10,626.43 g mol^−1^ and 8744.38 g mol^−1^ for recombinant CP12s with or without the 6xHis-Tag, respectively [42].

### 4.2. Samples Preparation

Reduction and oxidation of AtCP12 were performed together with the removal of the 6xHis tag [4,30,31]. For this purpose, half of the recombinant protein was incubated in the presence of 1 U of Thrombin protease (Cytiva) per 100 µg of pure AtCP12 in the presence of 20 mM 1,4-Dithiothreitol (reduced DTT), and half was incubated in the presence of Thrombin protease and trans-4,5-Dihydroxy-1,2-dithiane (oxidized DTT) at the same concentration as above. Both samples were incubated overnight at 22 °C.

To remove the 6xHis tags and the uncut form of AtCP12, a second nickel affinity chromatography (NiNTA resin; Cityva) was applied to the samples, collecting the unbound protein.

Reduced and oxidized AtCP12s were desalted in 25 mM K-phosphate buffer, pH 7.5, via a NAP-5 column (Cytiva) and brought to a concentration of 14.43 mg mL^−1^ and 14.00 mg mL^−1^, respectively, by ultrafiltration (Centricon YM3). Permeates of the ultrafiltration processes were preserved and used as reference in subsequent experiments. All samples were stored at −80 °C until use.

### 4.3. SAXS Data Collection

SAXS measurements were performed at the BioSAXS beamline BM29 of the European Synchrotron Radiation Facility (ESRF), Grenoble (France) [43]. Sample details and data collection parameters are also reported in Appendix A. After thawing, the samples were centrifuged and diluted to final concentrations between 5.7–0.25 mg mL^−1^. The SAXS measurements were performed with the automatic sample changer available at the beamline. The capillary and storage temperature were set at 4 °C. Sample volumes of 50 μL were withdrawn, and a set of 10 consecutive exposures was acquired during sample flowing in the capillary. The data reduction of 2D images (masking, azimuthal averaging and transmission scaling) was performed tvia the analysis pipeline [44] to obtain one-dimensional intensity profiles I(q), where q is the scattering vector modulus:q = 4πsin θ/λ(1)

Here, 2θ is the scattering angle, and λ is the X-ray wavelength. The I(q) profiles obtained from the frames were automatically compared to assess radiation damage and then averaged. The scattering contribution of the capillary filled with buffer was subtracted and the intensity was divided by the protein mass concentration. The absolute intensity scaling to macroscopic scattering cross section (I(q) in cm^−1^) using water as a standard, after normalization for concentration, provided intensities expressed in terms of protein molecular weight (kDa) assuming a protein-specific volume value of 0.735 cm^3^·g^−1^, which gives a squared scattering contrast per mass of protein ${∆\rho }_{M}^{2}$ of 5.04 × 10^20^ cm^2^·g^−2^:(2)$$MW[kDa]=\frac{10^{-6}\left[\frac{kDa}{Da}\frac{mg}{g}\right]I\left(q\right)\left[{cm}^{-1}\right]\cdot N_{A}[{mol}^{-1}]}{{∆\rho }_{M}^{2}\left[{{cm}^{2}\cdot g}^{-2}\right]\cdot c\left[mg\cdot {cm}^{-3}\right]}$$

Two repetitions of the measurement procedure for each protein concentration were run, and the data were averaged.

### 4.4. SAXS Data Analysis

Analysis of the scattering profiles was performed with the tools of ATSAS 3.2 [45]. The intensity was extrapolated at zero angle I(0), and the radii of gyration R_g_ were calculated for the data at all concentrations using the Guinier approximation:(3)$$\ln\left(I\right)=\ln\left(I\left(0\right)\right)-\frac{{R_{g}}^{2}}{3}q^{2}$$

Considering three different q ranges. The R_g_ values obtained considering data within 0.22 < q < 0.41 nm^−1^ were systematically higher compared to those obtained neglecting these first data points, which were comparable for the Guinier fit performed within ranges 0.36 < q < 0.59 nm^−1^ and 0.44 < q < 0.78 nm^−1^ (Appendix A). Therefore, it was assumed that the analysis of the experimental SAXS profiles starting from a minimum of q > 0.4 nm^−1^ can be considered devoid of the influence of small amounts of larger scattering objects or protein aggregates that could affect the scattering profiles in the very initial portion. Regarding the concentration dependence, the R_g_ values were observed to slightly decrease at high concentration (due to possible influence of particle–particle repulsive structure factor) and remain constant for concentrations below 2 mg mL^−1^. Since the noise level was rather high below 1 mg mL^−1^ (Appendix A), final data sets for the oxidized and reduced protein were obtained by merging the profile at 1 mg mL^−1^ concentration for q < 0.4 nm^−1^ and the profile at maximum concentration for q > 0.6 nm^−1^, considering the average of the two profiles in the overlap region, allowing for a scaling constant of 0.93 for the 1 mg mL^−1^ data due to possible errors in concentration determination. The final merged profiles were used for further analysis.

The indirect Fourier transform method was applied to obtain the P(r) function, with an estimate of the maximum particle dimension (D_max_), in addition to an independent calculation of I(0) and R_g_. The molecular weight was also estimated via concentration-independent methods and the tool based on Bayesian inference to estimate a most probable value, and a confidence interval was employed [46], even if some of the assumptions involved in these approaches could be less valid for disordered proteins.

### 4.5. Ensemble Fitting with Multiple Pools

The ensemble method implemented in EOM 3.0 [36,37] was applied to model the SAXS profiles of oxidized and reduced AtCP12 and deduce information about the size distributions and conformational flexibility of the protein in the two conditions. Pools of 10,000 conformers were generated according to different assumptions as explained in the Results section, using the RANCH algorithm. The theoretical SAXS profiles for all models of each pool were calculated using the FFMAXER tool, which uses the CRYSOL method, imposing default settings for the hydration layer and atomic excluded volume, 25 spherical harmonics and 301 data points. The genetic algorithm optimization implemented in GAJOE was employed to select from the pool ensembles of conformers for which averaged scattering profiles would fit the experimental data, allowing for background constant optimization. A fixed ensemble size of 50 curves per ensemble without repetitions was imposed according to recent recommendations [47] to avoid artifacts when applying the method to short proteins. Eight GAJOE runs (each involving 100 automatic repetitions) for each pool were run in order to estimate standard deviations for the average dimensional parameters (<R_g_> and <D_max_> of selected final ensembles and of overall generated histogram in the 100 repetitions), for the optimized constant background and for the discrepancy index between experimental and calculated SAXS profiles:(4)$$\chi^{2}=\sum_{i=1,\ldots N}{\left(\frac{I_{calc}\left(q_{i}\right)-I_{exp}\left(q_{i}\right)}{\sigma_{i}}\right)}^{2}$$
where N is the number of data points in the scattering profiles.

## Figures and Tables

**Figure 1 ijms-24-09308-f001:**
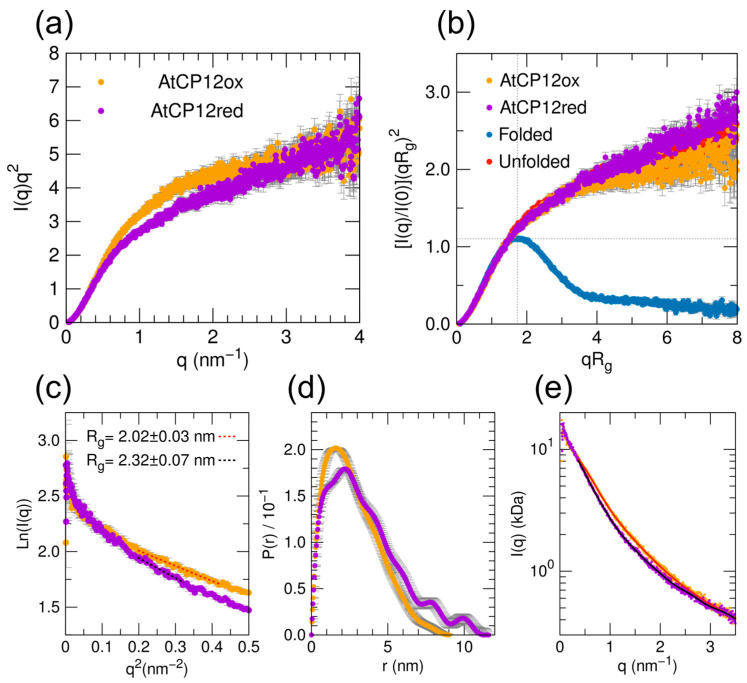
SAXS profiles of AtCP12 in oxidized (AtCP12ox) and reduced (AtCP12red) states. (**a**) Kratky plot of AtCP12 in oxidized form (orange dots) and reduced form (purple dots); (**b**) dimensionless Kratky plot of oxidized and reduced AtCP12 compared to a folded reference (BSA, blue dots, experimental) and an unfolded reference (SASBDB entry SASDNP6 [32], red dots); (**c**) Guinier plot with linear fit for determination of the radius of gyration R_g_ and of the intensity extrapolated at zero angle I(0); (**d**) pair distance distribution functions P(r) obtained by indirect Fourier transform and (**e**) corresponding fits to the data (red solid line for oxidized AtCP12 and black solid line for reduced AtCP12).

**Figure 2 ijms-24-09308-f002:**
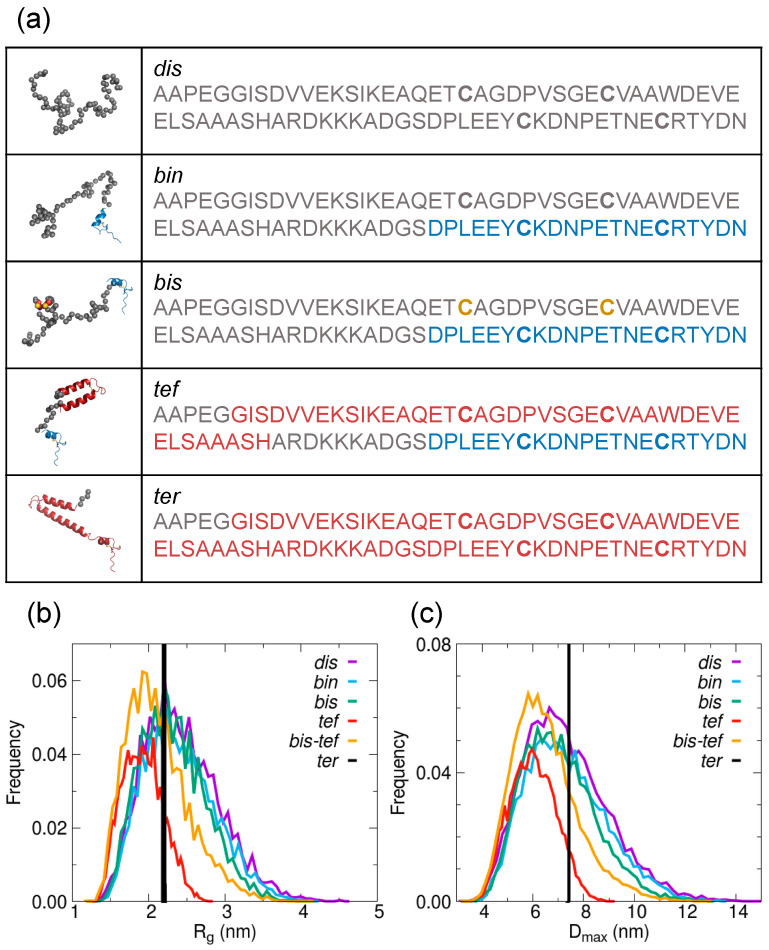
Graphical representation and structural analysis of the five conformer pools listed in ascending structural order generated to model the SAXS data of AtCP12 in oxidized and reduced form with the software EOM. (**a**) One example of the generated conformers (10,000) is shown for each AtCP12 pool (left): the dummy residues described as a disordered chain are shown as grey spheres, whereas the structured portions described according to the AtCP12 coordinates from the crystallographic structures PDB ID 3QV1 (chain G) [38] for *bin* and *bis* pools and PDB ID 6KEZ (chain N) [25] for *tef* and *ter* pools are shown in pink (N-terminal) or blue (C-terminal); in pool *bis*, Cys22 and Cys31, which were involved in the N-terminal disulfide for which a distance constraint was imposed, are shown as colored spheres. The AtCP12 sequence (right) is colored with the same color code used for the corresponding models. (**b**) R_g_ and (**c**) D_max_ distributions for the different pools as summarized in (**a**) and for the additional pool comprising both conformers of the *bis* and *tef* pools (20,000 models, *bis-tef*).

**Figure 3 ijms-24-09308-f003:**
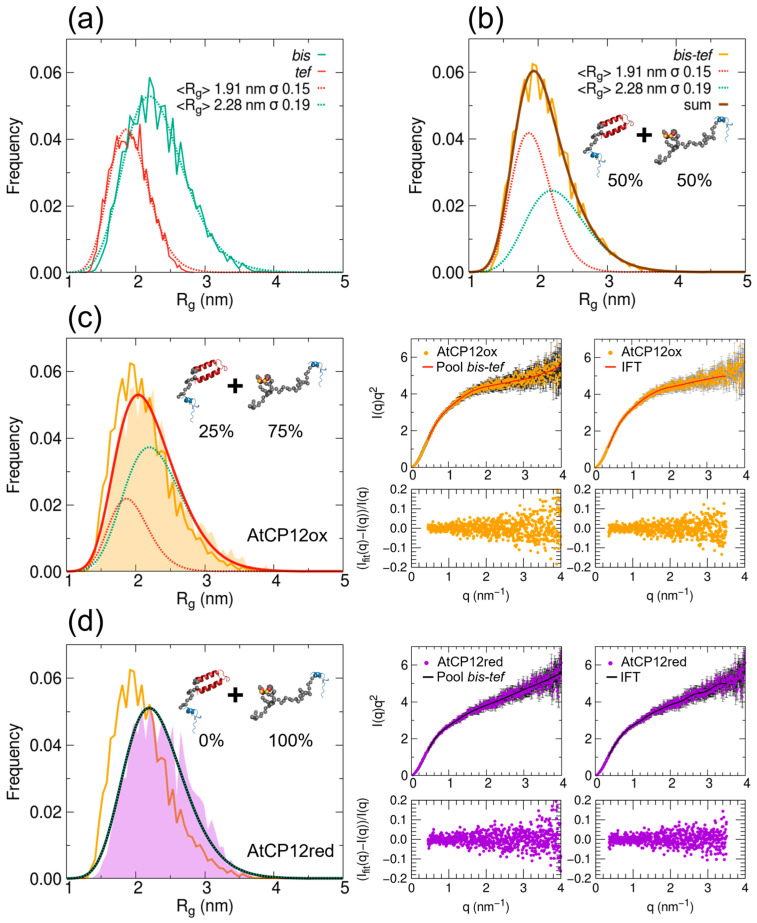
Results of ensemble optimization for oxidized and reduced AtCP12 considering a joint conformer pool with both disordered (*bis*) and structured (*tef*) N-terminal region. (**a**) Comparison between R_g_ distributions of the two individual pools (*bis* in green and *tef* in red) and fit with log-normal functions. (**b**) R_g_ distribution of the joint 50% *bis*–50% *tef* pool of conformers and fit as a sum of the two underlying distributions corresponding to the individual pools. (**c**) Left panel: R_g_ frequency distributions for the initial pool (orange solid line) and for the final selected ensembles to fit SAXS data of oxidized AtCP12 (orange shaded area), which is fit (red solid line) as a sum of two underlying distributions corresponding to the individual pools with adjusted weights (green and red dotted lines); example models from the two pools and the adjusted weights are shown. Right panels: theoretical SAXS profile (solid line) superimposed to the experimental data points in Kratky plot representation (dots). Below, the corresponding normalized residuals (I_calc_(q) − I_exp_ (q))/I_exp_ (q) are shown. As a reference for a good fit, the agreement obtained by indirect Fourier transform is shown on the right. (**d**) Same as (**c**) for AtCP12 in reduced form, with the final selected ensembles to fit SAXS data of reduced AtCP12 (purple shaded area), which is fit (black solid line) as a distribution corresponding to the *bis* pool. The result of the optimization with the lowest χ^2^ among the eight repetitions is shown in both cases.

## Data Availability

The SAXS data of AtCP12 in oxidized and reduced form are available on Small Angle Scattering Biological Data Bank (SASBDB) with the following accession codes: SASDSM2 for oxidized AtCP12 and SASDSN2 for reduced AtCP12.

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
