# Peer review of "Conformational Disorder Analysis of the Conditionally Disordered Protein CP12 from Arabidopsis thaliana in Its Different Redox States"

_ijms, 2023, doi:10.3390/ijms24119308_

Round 1

Reviewer 1 Report

I’m well aware that analyzing the disorder structures is one of the most challenging tasks in structural biology. In this paper, the authors capture structural differences in the presence or absence of disulfide bonds in Arabidopsis thaliana CP12 by small-angle X-ray scattering analysis. Using the EOM method, they were able to derive the ratio of structure groups from SAXS data containing contribution from different conformations. The authors also touch upon previous studies and mention the difference between the previously reported structural properties of Chlamydomonas CP12 and those of Arabidopsis CP12. While the findings are not significantly divergent from previously predicted structural features, their experimental confirmation is noteworthy and deserves recognition.

In the ‘Results’ section, the distribution of the obtained plots is explained in detail with diagrams and graphs. While this may be somewhat redundant for SAXS scientists, it is appropriate for many readers of this journal, including myself, who are not proficient in SAXS analysis.

Line 310 TFE should be described as 2,2,2-trifluoroethanol without abbreviation.

Line 309-311 The effects of TFE against proteins are mediated not only by a reduction in solvent polarity but also by clustering of alcohol (D. P. Hong et al., J. Am. Chem. Soc., 1999, R. M. Culik et al., J Phys Chem B. 2014). I think that the condition in presence of TFE cannot easily be equated with protein-protein interaction environment.

Line 347-352 One of the most important aspects of this paper is that the oxidized and reduced forms of CP12 used in the SAXS measurements are indeed samples with different cysteine redox states. In ‘Materials and Methods’ section, although the addition of oxidized and reduced forms of DTT was mentioned, the cysteine redox state of the obtained proteins was not experimentally verified. The authors need to perform the quantification of free cysteine of them to show that the redox state is as expected. Alternatively, it is necessary to refer to previous report to show that the redox state of cysteine residues in CP12 can be controlled by similar experimental procedure.

Line 455-456 The author’s SAXS data has not been submitted to the database. In the protein crystallography, it is common to obtain a Protein Data Bank ID before submitting a manuscript. I do not fully understand the SASBDB deposition system, so there may be some misunderstandings, however confirm the general procedure.

Author Response

Reviewer 1

I’m well aware that analyzing the disorder structures is one of the most challenging tasks in structural biology. In this paper, the authors capture structural differences in the presence or absence of disulfide bonds in Arabidopsis thaliana CP12 by small-angle X-ray scattering analysis. Using the EOM method, they were able to derive the ratio of structure groups from SAXS data containing contribution from different conformations. The authors also touch upon previous studies and mention the difference between the previously reported structural properties of Chlamydomonas CP12 and those of Arabidopsis CP12. While the findings are not significantly divergent from previously predicted structural features, their experimental confirmation is noteworthy and deserves recognition.

In the ‘Results’ section, the distribution of the obtained plots is explained in detail with diagrams and graphs. While this may be somewhat redundant for SAXS scientists, it is appropriate for many readers of this journal, including myself, who are not proficient in SAXS analysis.

QUESTION: Line 310 TFE should be described as 2,2,2-trifluoroethanol without abbreviation.

ANSWER: DONE

QUESTION: Line 309-311 The effects of TFE against proteins are mediated not only by a reduction in solvent polarity but also by clustering of alcohol (D. P. Hong et al., J. Am. Chem. Soc., 1999, R. M. Culik et al., J Phys Chem B. 2014). I think that the condition in the presence of TFE cannot easily be equated with protein-protein interaction environment.

ANSWER: We thank the Reviewer for his/her comment. We modified the sentence by removing the statement about the use of the TFE as a proxy for protein-protein interaction, and rather added a reference about the use of TFE in the study of the secondary-structure propensity in proteins.

QUESTION: Line 347-352 One of the most important aspects of this paper is that the oxidized and reduced forms of CP12 used in the SAXS measurements are indeed samples with different cysteine redox states. In ‘Materials and Methods’ section, although the addition of oxidized and reduced forms of DTT was mentioned, the cysteine redox state of the obtained proteins was not experimentally verified. The authors need to perform the quantification of free cysteine of them to show that the redox state is as expected. Alternatively, it is necessary to refer to previous report to show that the redox state of cysteine residues in CP12 can be controlled by similar experimental procedure.

ANSWER: We thank the Reviewer for its very appropriate comment. The redox state of AtCP12 cysteines was deeply investigated in previous studies and reported in already published papers. In Marri et al. 2010, for example, free cysteine thiols were determined by DNTB (Ellman) reagent both in reduced and oxidized AtCP12 obtained with the same protocol and a number of free thiols of 4.5 ± 0.5 and -0.2 ± 0.1 in reduced and oxidized condition, respectively, was reported. Based on these results, we are confident that the AtCP12 samples analysed by SAXS in this research, were effectively under its oxidized and reduced state. For this reason, we added in the Results and Material and Methods sections of the manuscript the references documenting the redox state of AtCP12 following DTT treatments and we apologize for the omission in the submitted version.

QUESTION: Line 455-456 The author’s SAXS data has not been submitted to the database. In the protein crystallography, it is common to obtain a Protein Data Bank ID before submitting a manuscript. I do not fully understand the SASBDB deposition system, so there may be some misunderstandings, however confirm the general procedure.

ANSWER: The sentence in the manuscript “The SAXS data of AtCP12 in oxidized and reduced form will be submitted to the Small Angle Scattering Biological Data Bank (https://www.sasbdb.org/” was indeed added before starting the actual submission procedure to SASBDB. This was done on 2023/05/12, the data for AtCP12 in oxidized and reduced form have been given draft ID 4955 and 4959, respectively and at the current date we still do not have the SASBDB code. In any case, we suggest leaving the following sentence “The SAXS data of AtCP12 in oxidized and reduced form have been submitted to the Small Angle Scattering Biological Data Bank (https://www.sasbdb.org/) under project name “Conformational disorder analysis of the conditionally disordered protein CP12 from Arabidopsis thaliana in its different redox states.” so that the data could be more easily traceable.

Reviewer 2 Report

This paper presents an analysis of the conformation of CP12 protein from Arabidopsis, when free in solution, in oxidized and reduced forms, using the tools of SAXS and EOM, and making a comparison with the CP12 protein from Chlamydomonas. The methods used are clearly described and are appropriate to obtain the desired information.

The conclusions drawn are: 

1) The reduced form of CP12 is essentially fully disordered, while the oxidized form appears to be a mix of fully disordered and partially ordered forms.

2) The fraction of partially ordered CP12 in the oxidized state is lower for Arabidopsis than for Chlamydomonas.

3) The more highly ordered conformations seen in complexes of CP12 with partner proteins must be induced during formation of the complexes.

The conclusions are justified based on the reported experiments, and the information may be of use in further study of systems involving CP12.

The work is well presented and the English is very good. A few minor comments:

Figure 1: The inset in panel (d) is so small that it's really impossible to see how well the points fit the line.

Figure 2: The caption mentions "crystallographic coordinates indicated in parenthesis", but I don't see any parentheses.

Line 60: "finely" should probably be "been firmly".

Line 115: "SASBDB" should be "SASBDB entry SASDNP6".

Line 265: "Differently" would be better as "In contrast".

Line 290: "5 residues highly dynamic" should be "5 highly dynamic residues".

Line 376: "Being 2theta the" should be "2theta is".

Author Response

This paper presents an analysis of the conformation of CP12 protein from Arabidopsis, when free in solution, in oxidized and reduced forms, using the tools of SAXS and EOM, and making a comparison with the CP12 protein from Chlamydomonas. The methods used are clearly described and are appropriate to obtain the desired information.

The conclusions drawn are: 

1) The reduced form of CP12 is essentially fully disordered, while the oxidized form appears to be a mix of fully disordered and partially ordered forms.

2) The fraction of partially ordered CP12 in the oxidized state is lower for Arabidopsis than for Chlamydomonas.

3) The more highly ordered conformations seen in complexes of CP12 with partner proteins must be induced during formation of the complexes.

The conclusions are justified based on the reported experiments, and the information may be of use in further study of systems involving CP12.

The work is well presented and the English is very good. A few minor comments:

QUESTION: Figure 1: The inset in panel (d) is so small that it's really impossible to see how well the points fit the line.

ANSWER: We thank the Reviewer and following his/her suggestion we removed the inset from panel (d) and turned it into panel (e). The figure legend has been changed accordingly.

QUESTION: Figure 2: The caption mentions "crystallographic coordinates indicated in parenthesis", but I don't see any parentheses.

ANSWER: We thank the Reviewer and we corrected the mistake due to an older version of the Figure panel, adding the pdb codes of the crystallographic structures considered in pool preparation in the Figure caption.

QUESTION: Line 60: "finely" should probably be "been firmly".

ANSWER: DONE

QUESTION: Line 115: "SASBDB" should be "SASBDB entry SASDNP6".

ANSWER: DONE

QUESTION: Line 265: "Differently" would be better as "In contrast".

ANSWER: DONE

QUESTION: Line 290: "5 residues highly dynamic" should be "5 highly dynamic residues".

ANSWER: DONE

QUESTION: Line 376: "Being 2theta the" should be "2theta is".

ANSWER: DONE

Round 2

Reviewer 1 Report

All my questions and comments were answered perfectly.

The quality of new manuscript is at a high level and deserves to be published.